# What emotions does music express? Structure of affect terms in music using iterative crowdsourcing paradigm

**Tuomas Eerola**[1☉]*, **Pasi Saari**[2☉]

**1** Department of Music, Durham University, Durham, United Kingdom, **2** Department of Music, Arts and Culture, University of Jyväskylä, Jyväskylä, Finland

☉ These authors contributed equally to this work.

* tuomas.eerola@durham.ac.uk

**Data Availability Statement:** All data is held in a public repository at OSF: https://osf.io/ve6gt/.

**Funding:** P.S. was supported by funding from ESRC (ES/K00753X/1). The funder had no role in

## Abstract

Music is assumed to express a wide range of emotions. The vocabulary and structure of affects are typically explored without the context of music in which music is experienced, leading to abstract notions about what affects music may express. In a series of three experiments utilising three separate and iterative association tasks including a contextualisation with typical activities associated with specific music and affect terms, we identified the plausible affect terms and structures to capture the wide range of emotions expressed by music. The first experiment produced a list of frequently nominated affect terms (88 out of 647 candidates), and the second experiment established and confirmed multiple factor structures, ranging from 21, to 14, and 7 dimensions. The third experiment compared the terms with external datasets looking at discrete emotions and emotion dimensions, which verified the 7-factor structure and identified a compact 4-factor structure. These structures of affects expressed by music did not conform to music-induced emotion structures, nor could they be explained by basic emotions or affective circumplex. The established affect structures were largely positive and contained concepts such as "romantic" and "free", and terms such as "in love", "dreamy", and "festive" that have rarely featured in past research.

## Introduction

People spend a great deal of time and energy on music. In the UK alone, music is estimated to contribute £6.7 billion to the UK economy [1]. One of the main reasons people typically explain their involvement with music is to engage with emotions [2]. The emotions that music can *induce* in listeners have been an intense topic of research over the past two decades, and there are several widely used accounts of these [3, 4]. However, the emotions that music considers to *express* and *communicate*, which is a distinctly different process and topic [5], have received considerably less attention than the emotions induced by music. The most comprehensive current characterisation of the emotions people think music can express is from 2004 [6], which suggested that music most frequently can express joy (99%), sadness (91%), love

study design, data collection and analysis, decision to publish, or preparation of the manuscript.

**Competing interests:** The authors have declared that no competing interests exist.

(90%), calm (87%), anger (82%), and so on. This list of 39 emotions did not contain any feasible structure of emotions expressed by music [see also [7]].

In addition to these surveys, music information retrieval research that has focused on solving the music emotion recognition (MER) challenge has focused on valence and arousal [8–14], or basic emotions [15, 16]. The notable exceptions are to these are emotion taxonomies proposed after analysis of a large number of mood tags that offered five clusters of emotional expression [17] which has been used in the MIREX audio mood classification task [18, 19], and similar analysis mood tags that produced variant structures of expressed affects and affective circumplex solution from the tags [20]. An update to the taxonomies of emotion terms used since the 1930's [21] and 1950's [22] was proposed that consisted of 46 terms [23]. In short, our understanding of the richness and structures of emotional expression in music is not at the level of contemporary research on emotions induced by music. This limits our possibilities of capitalising on the richness of emotions expressed by music in applications (emotion recommendation, music generation, and music therapy).

Since the two surveys conducted in Sweden [6, 7], the update by Schubert [23] of the adjectives describing music defined by Hevner [21] and Farnsworth [22], the tag data using AllMusicGuide.com, experts, and usage data [18], there is no contemporary, data-driven account of the expressed emotion vocabulary or the expressed emotion structure for music. It should be stressed that the discovery of emotion induction structures such as *Geneva Emotion Musical Scale* [3] or other variants cannot be considered adequate, as music is believed to be capable of expressing a more diverse range of emotions. Most emotion induction models neglect the full extent of emotions expressed by music, with significant expressive terms such as "love", "longing", and "humor" frequently absent. Moreover, empirical discovery of the emotions expressed—and even the emotions induced in listeners—has neglected to address the full contextualization of music in its many uses for daily activities and functions. [24, 25], nor have prior studies conducted experiments in conjunction with participants listening to actual music to elucidate the structure of emotion expressed in the music.

## Structure of emotions expressed by music

In affective science, multiple structures of affects have been proposed. The basic emotion framework [26, 27] has been a popular choice for emotions, although the number of emotion categories has not been agreed upon [28–30]. Another common framework for emotions, both for expression and for induction, is a set of small dimensions, usually valence and arousal, as proposed by Russell (1980). When this affective circumplex is used to describe expressed emotions in music, the four quadrants are used to define the valence-arousal space [8, 31], or the way affect terms are loaded into this 2-dimensional place is capitalised [32]. It is worth noting that a mapping of emotion cues from music with the basic emotion terms tend to yield a less consistent picture than when those cues are mapped against the quadrants in the affective circumplex [33].

In the case of emotions induced by music, a large-scale survey of affect terms by Zentner et al. [3] established a 9-dimensional structure of emotions induced by music (wonder, transcendence, tenderness, nostalgia, peacefulness, power, joyful activation, tension, and sadness), which could be refined into 45-affect clusters and three meta-factors (sublimity, vitality, and unease). Their work touched upon the emotions expressed by music, but unfortunately, only reported broad differences between induced and expressed emotions. Recent variations of this model have been proposed, namely GEMIAC (14-dimensions, [34]), AESTHEMOS (21-dimensions, [4]), and the 13-dimensional model by Cowen, Fang, Sauter, and Keltner [35]. Although the structure of these models partially overlaps, the diversity also suggests that

the variation in induced emotions may stem from various sources, from the backgrounds of participants, to the genres they were mostly familiar with, or the type of terms that were initially used in these surveys.

Outside of music, analyses of affect terms have initially suggested a clear low-level (valence and arousal, some basic emotions) organisation of affects [36], and large-scale cross-cultural analyses of affect terms in thousands of languages using colexification analyses [37] have suggested that the fundamental affect structure may only support the dimensions of valence and arousal of affect terms. The precise connotations of similar words tend to reflect cultural practices and regional variations. When the emotional expression of art products has been explored, researchers have adopted a less structural approach and asked artists and audiences how well the artworks communicate a specific list of 37 affect terms [38], suggesting that some of the emotional expressions can be communicated through the art works, but the variability is large. From these summaries, it is clear that while music-induced emotions have been reduced down to a range of 9 to 21 dimensions of emotions, the structure of emotions expressed by music has not received similar attention; currently, we have only ranked lists of affect terms [6], five mood clusters [17], or affect words organised around the valence-arousal space [21, 23, 32] as a starting point for capturing the emotions expressed by music.

## Research question

Our primary research question is what emotions are expressed by music? What are the relevant terms and structures encapsulated by the terms? There has not been a direct study of the terms and structure of the expressed emotions of music in 20 years [6], and it is time to carry out a series of studies with additional validations with external data to establish a feasible structure of emotions relevant for music in the western context. In contrast to past efforts, we want to carry this mapping in a context where we ask for contextual information about affects and allow the participants to select and hear actual musical examples as part of the decision-making process. For this purpose, we have designed a new iterative paradigm.

**Iterative paradigm.** A custom paradigm was created, where participants can select appropriate affect terms, tracks, and activities that go with terms. This paradigm/task will iteratively, across participants, produce information about the appropriate activities associated with music, the relevant affect terms, and how well the affect terms are communicated across participants. As people's use of music in everyday life is highly contextual and functional [24, 25, 39], we wanted to remind participants of typical, concrete, and practical contexts in which music is listened to and to connect these contexts with affect terms. For this reason, we asked participants to explicitly link each affect term with a selection of nine activities, which cover a vast majority of activities and functions people typically engage with music [24, 25, 39] and ask participants to explicitly link each affect term with the relevant activities. Contextualising the relationship between the affect terms and the music has a purpose in that it broadens the participants mindset to consider the boundaries of what music can fully express. Finally, we also want participants to be able to connect the affect terms with the actual music, allowing us to bridge the gap between what affects the music *could* express and what affects the music can successfully communicate.

Our paradigm has three subtasks: (1) activity association, (2) track submission, and (3) track annotation (see Fig 1). We next describe the subtasks in detail.

**Activity Association** In the activity association subtask, participants were asked to associate the affects expressed by music with different types of daily activities, listed in Table 1. A single affect term was presented and the participant was asked to check all activities that would fit the music expressing the emotion. An activity checklist was presented below the affect

**Fig 1. Three subtasks illustrated.** *Level a* refers to the input to the participant, which in the activity association is a single affect term, in track submission 24 terms from which they selected one, and in track annotation task, it was a music track proposed earlier by another participant to represent an affect term. *Level b* refers to the task where they either associated an affect term with an activity (task 1), or searched a music track representing their chosen affect term (task 2), or chose the best matching affect term for a track chosen by another person (task 3). The selection of affect terms was carried out according to $x_1 \ldots x_N \in_R S$ where $S$ is a set of 647 affect terms from which $N$ were randomly chosen ($N = 24$ in the activity association task and $N = 22$ in the track submission task). The nine activities were always presented in full with definitions.

term, and the participant was required to check at least one box. In addition to the activities, an option was given to check a box "Not relevant mood for music", which would uncheck all other boxes, or a box "Unfamiliar mood term", which would lead to skipping the affect term altogether. Furthermore, it was possible to check the dictionary definition of the affect term in case the participant was unsure of the meaning of the term.

**Track Submission** Track submission subtask involved three stages: affect term selection, track submission, and activity annotation. In the selection of affect terms, a list of affect terms

**Table 1. Summary of activities.**

| Activity | Examples | Description |
|---|---|---|
| On the move | driving, walking, transport | Moving from one place to another by any means (e.g., car, bike, foot, plane, or public transport) |
| Daily routines | washing, cleaning, cooking | Domestic chores and everyday tasks |
| Intellectual | studying, reading, writing | Brain work, such as private study and desk work at occupational settings |
| Entertainment | tv, internet, games | Using primarily non-musical media for entertainment |
| Physical | dancing, sports, relaxing | Physical activities such as exercise, relaxation, and pain management |
| Emotional | reminiscing, meditating, charging | Mental activities whose purpose is to manage mood, self, and emotions |
| Live music | concert, gig, performance | Participating to activities involving live music as an audience member or performer |
| Social | gatherings, dining, night out | Any activity involving a social aspect—both informal and formal |
| Music listening | focussing on music | Music listening as the main activity |

was presented, and the participant was asked to select a term for which a music example expressing that affect could later be associated with. In the track submission stage, an online commercial music catalogue (7digital) was searched for tracks that expressed the selected affect. A free text search based on track title, artist or album returned a list of tracks out of which the participant could listen to excerpts from the tracks (30-60 s). The desired track was then submitted. In the activity annotation stage, a checklist similar to that of the activity association subtask was presented. The participant was asked to check all activities that fit the submitted track expressing the selected affect.

**Track Annotation** The track annotation subtask involved two stages: affect annotation and activity annotation. An example of music, selected from the tracks already submitted in the track submission subtask, was played to the participant. The track was selected from the tracks that had not been submitted by the participant herself. A checklist of affect terms was presented, out of which zero or more terms could be checked that were expressed by the music example in the participant's opinion. In the activity annotation stage, the participant was asked to check all activities that fit the music example.

## Rationale and overview

To establish what relevant affect terms are and the structures for emotional expression of music, we used an iterative paradigm in three experiments; In Experiment 1, we will explore a large number of affect terms and trim the selection to a consistently used and relevant terms. In Experiment 2, we will analyse the structure within the affect terms through a series of exploratory and confirmatory factor analyses. In Experiment 3, we will validate the structure and make comparisons with existing datasets annotated with well-known concepts such as basic emotions and valence and arousal.

## Experiment 1

The aim of Experiment 1 was to assess the relevance of a large number of affect terms to music listening, and to cluster the affect terms according to their similarity to make the list more concise while retaining the variability of terms.

## Methods

Ethics approval (for all experiments reported in this manuscript) was obtained from the institutional committee. Participants were provided with informed consent on the survey landing page, which provided information about the study, the voluntary nature of their participation, the risks and benefits of participation, and anonymity, and an option to choose "I Agree" and "I Do Not Agree" to give their consent. Data collection was carried out between 1/4/2015 and 30/6/2015.

**Materials.** To cover a wide variety of affect terms relevant to music, we follow the approach previously adopted in [3, 32] and aggregated affect term lists from several sources such as research articles that focus on emotions expressed [6, 40] and induced by music [3, 6, 40], sources from general affective sciences [36, 41, 42], as well as online music services that index songs by emotion (*Allmusic*, *Audiosparx*, *Sonoton*). Aggregating the obtained lists and merging synonyms, inflicted forms, and different word classes of the same term resulted in a list of 647 affect terms; see Original terms in S1 File. After this, an adjective form was sought for all terms. The language check of the resulting list was performed by a native English speaker.

To contextualise the affect terms and the choices in various actual uses of music, we also asked about activities that could go along with music or in life in general. The importance of the everyday activities was collected on a scale of *1 = Not at all—5 = Extremely*. Nine activities were used that were identified in previous studies as important contexts of everyday life for listening to music [24] (see Table 1). Additionally, from all participants we collected basic demographic information (age, gender, country), English language proficiency, musical expertise [43], and a measure of musical preference from an adapted 16-genre version of the *Short Test of Musical Preferences* [44].

**Participants.** While we focus on western music, employing English language and those using computers in the experiment, we wanted to allow a diverse range of people from different continents and countries to participate in the experiment. Participants were recruited from a crowdsourcing platform CrowdFlower.com, which allows the creation of human annotation tasks that are distributed across its registered participant pool. A total of 1857 participants participated in Experiment 1. The age distribution (M = 32.04, SD = 10.03) of the participants resembled that of MTurk [45], 72.59% reported at least a limited working proficiency with English, and 71.3% were males. Non-musicians or music-loving nonmusician were the most frequently reported level of musical expertise (77.05%). Regarding the inferred geographical locations of the participants, the majority came from Europe (46.6%), Asia (28.2%), South America (12.3%), or North America (9.8%), for more details, see S1 Table in S1 File. According to the musical genre preference survey, participants reported to prefer pop, rock, classical, electronic, and hip-hop (all mentioned by at least by 25% of participants). A complete breakdown of the preferred genres is given in S2 Table in S1 File. The most important activities were entertainment (M = 3.88, SD = 1.00), music listening (M = 3.84, SD = 1.06), and commuting (on the move) (M = 3.73, SD = 1.01), and intellectual (M = 3.57, SD = 1.16) while the four main activities with music listening were music listening (M = 4.16, SD = 1.04), commuting (on the move) (M = 3.80, SD = 1.09), physical (M = 3.73, SD = 1.09), and live music (M = 3.71, SD = 1.24), see S3 Table in S1 File for complete breakdown.

**Procedure.** Data acquisition was done on a separate web page from the crowd-sourcing platform to allow flexible setup of the annotation. The interface was optimised for mobile and desktop displays. The recruited participants logged into the interface using their Crowdflower username so that they could be identified after they logged in again. After completing a series of subtasks, constituting one task, the participants received a token that they typed on the crowd-sourcing platform to receive the reward payment. Payment was set at 10 cents for each completed task. No limits were imposed on the number of tasks for each participant.

The tasks were presented in a random order, and each participant had to carry out a minimum of 10 tasks (6 activity associations, 2 track submissions, 2 track annotations). Since we have a large number of affect terms, we presented a random selection of terms to each participant according to $x_1 \ldots x_N \in_R S$, where $S$ is a set of 647 terms and we randomly choose $N$ terms in different tasks; for activity association, $N=1$, for track submission task, $N = 24$; and in the track annotation, the same 24 affect terms were presented that were used for the same musical example in the track submission task. The intention was to collect approximately 50 responses for each affect term in the Activity association, which would correspond to approximately 5500 completed tasks in total.

## Results

In Experiment 1, 1857 participants submitted 4472 unique tracks (mean duration 4 min 24s, 2227 unique artists, 3465 unique releases). The artists with most of the tracks were Eminem (42), Metallica (33), and Michael Jackson (33). The tracks submitted the most often were John

Legend: All of Me (44), Pharrell Williams: Happy (from "Despicable Me 2") (42), and Avicii: The Nights (42). A median of 50 activity associations were collected for each affect term, each affect term was selected 8 times (median) in the track submission subtask, and each affect term was tagged 24 times (median) in the track annotation subtask. The affect term "happy" was associated with one or more activities the most frequently (37%), "romantic" was the most frequently selected term for track submission (26% frequency), and "rhythmical" was the most frequently applied term in the track annotation task (27%).

Based on the collected data, the affect terms were eliminated and clustered based on the following relevance criteria:

(1). *Familiarity*: Terms marked as unfamiliar by at least 20% of the participants were eliminated. These terms were considered unidentifiable to a large proportion of the general population (based on activity association). (2) * Activity relevance: * Terms that did not fit any activity according to at least 20% of the participants were eliminated (based on activity association).

(2). *Associability to music tracks*: Terms used for the track search less often than what the chance rate indicates were eliminated (based on track submission). The chance rate was counted as the number of completed track submission subtasks divided by the number of affect terms used in the task (24). This filtered out affect terms that could not easily be associated with music tracks.

(3). *Consistency*: Terms selected in the track submission subtask but never tagged on the same track were eliminated (based on track submission and annotation). This filtered out affects expressed by music that were not frequently agreed upon between the two participants who submitted and annotated a musical example.

(4). *Activity similarity*: Terms that fit the same set of activities were identified by clustering (based on activity association), and the term most frequently associated with music tracks was retained for each cluster (based on track submission and annotation).

Applying the familiarity criteria (1) led to elimination of 6 terms (e.g. "quirky", "organic", "nihilistic", "messy", "schmaltzy", and "glum"), activity relevance (2) eliminated further 81 terms (e.g. "gay", "crunk", "creepy", "sick", "technical", "wicked", "scary", "terrified", "sleazy", and "hateful"), associability (3) to music tracks eliminated 336 terms (e.g. "trancelike", "swinging", "droning", "ethereal", "summery", "trippy", "abstract", "horrified", "bittersweet", and "wintry"), and consistency (4) criteria eliminated 11 terms (e.g. "mellow", "humorous", "fantasy-like", "raging", "complex", "regretful", "curious", "shocked", "surprised", and "infatuated"). This resulted in a list of 213 affect terms. To reduce the number of unique terms more, we explored similarities between the affect terms by using k-means clustering, which was based on a term-activity distance matrix. To determine an optimal number of clusters, we used the silhouette score [46]. This analysis yielded the maximal score with 88 clusters, which was then applied as the *K* for the k-means clustering to reduce the number of affect terms, from 213 to 88.

Fig 2 shows the properties of the affect terms. Panel A displays the search frequency of the top 213 affect terms. The ones shown in grey are those trimmed through the process described earlier. Panel B shows the distances between the 88 affect term activity nominations and the potential structure inherent in these. The obtained clusters were in general semantically coherent, such as Moody ("unhappy", "sorrowful", "hopeless", and "depressed"), and Liberated ("fierce", "raw", "pissed off"), although some clusters were more coherent with regard to activity fit, such as Slow ("suggestive", "heartwarming", "innocent", "warm", "spirited", "magical")

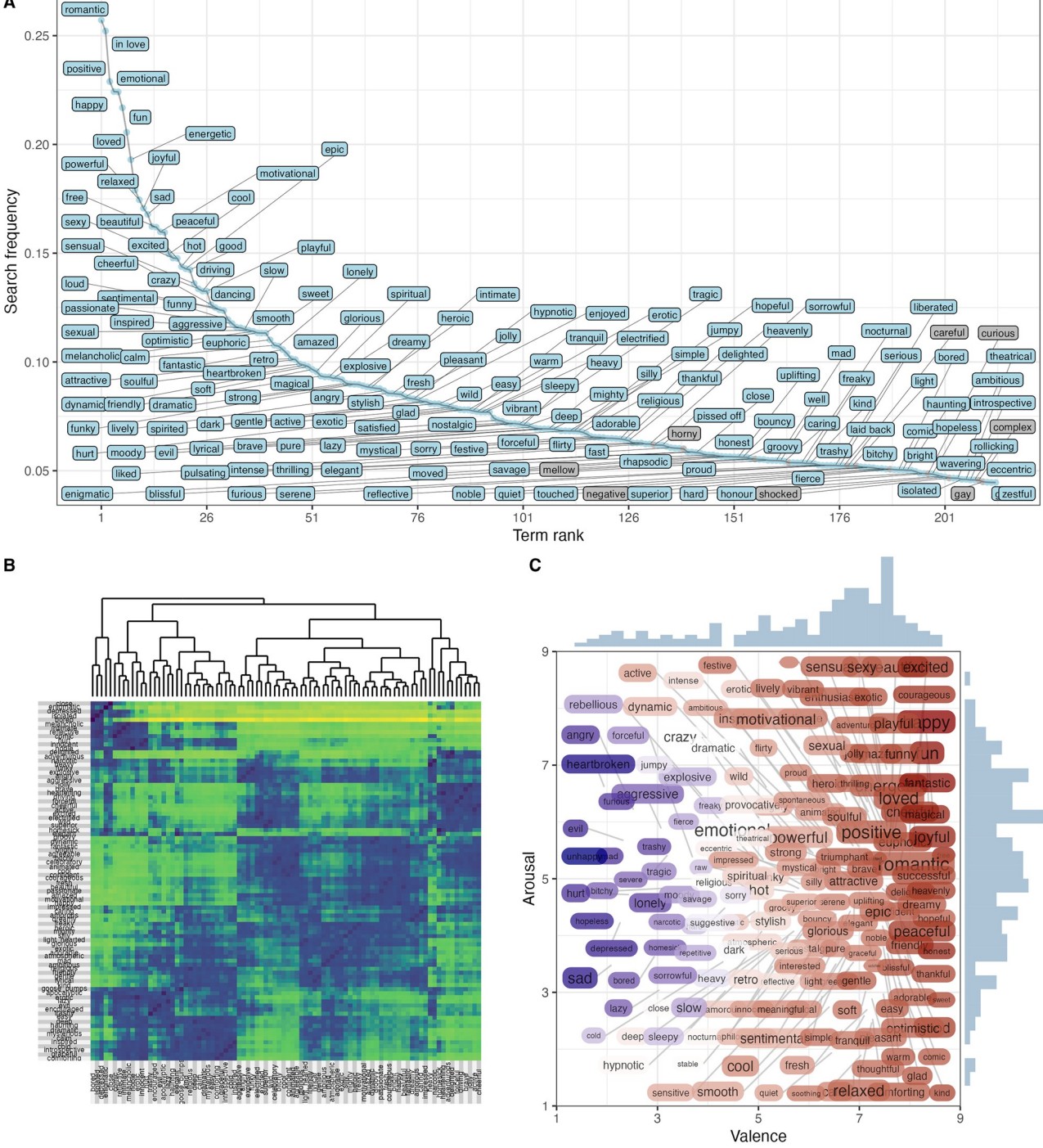

**Fig 2.** Visualisation of (A) search frequency of 213 affect terms, (B) term similarities and clustering solution for 88 prominent terms, and (C) the 88 terms positioned within the affective circumplex space—obtained by projecting the valence and arousal values for each term by Warriner and Brysbaert [47].

and Sorry ("sorry", "heavenly", "mysterious"). Many clusters consisted of single affect terms such as "pulsated", "inspired", "powerful", and "satisfied". Panel C plots the affective circumplex space covered by the 88 terms, obtained by projecting the valence and arousal values from Warriner and Brysbaert [47]. The size of the term reflects the search frequency. What is immediately clear from the distribution of affect terms in the affective space is that the majority (64%) of the terms are on the positive side of the valence axis (median valence of 6.7 on a scale of 1-9), but the arousal dimension is balanced more evenly (median of 4.8).

## Discussion

At this point, we have identified a candidate list of 213 relevant affect terms that may be captured with 88 term clusters. We do not wish to embark on more analytical comparisons of this structure to existing emotion structures yet, as more data would make structure discovery more robust and the purpose of this stage was to compress the terms into a representative and music-associable set. Suffice to say, many but not all of the top-ranking terms in our data can be observed in the past survey results of expressed emotions. For example, "joy", "sadness", "love", "calm", "anger", and "tenderness" can be found in the top 5 of the Juslin and Laukka study [6], but their main terms or their variants have ranks of 11, 12, 2, 42, 68, and 282 in the search frequency in our data. Consequently, many of the top terms in our data, such as "romantic" (1), "positive" (3), "fun" (6), or "energetic" (8) do not exist in previous surveys [6, 7]. As this could be related to the choice of synonyms, translations, and broader underlying structures of affects, we next pursue the broader picture by engaging in a new round of data collection across all three subtasks but using a smaller set of affects (88) identified here.

## Experiment 2

The primary objective of Experiment 2 was to examine the structure of the affects expressed by music to discover a factor model that would represent a wide range of musical genres and listener demographics. The same paradigm was used with three subtasks, but using the 88 terms suggested in Experiment 1. Exploratory (EFA) and confirmatory (CFA) factor analyses were applied to the data obtained.

### Methods

**Materials.**   Those 88 affect terms that were retained in Experiment 1 were included in the materials. Furthermore, the tracks from Experiment 1 that had been submitted for one of the 88 retained mood terms in the track submission subtask were included, as well as all the responses from all the subtasks from Experiment 1 related to the retained mood terms and tracks.

**Participants.**   A total of 2508 people participated. Demographics were similar to those of Experiment 1; Mean age 32.01 (SD = 10.19), 71.70% reported limited working proficiency or higher in language skills and 71.2% were men. Non-musicians or non-musicians who love music were the common level of musical expertise (73.89%). Regarding the inferred geographical locations of the participants, most of the participants were from Europe (48.0%), Asia (28.6%) or South America (11.7%) and North America (8.9%).

**Procedure.**   Experiment 2 utilized the same three subtasks described in Experiment 1. Each participant did a minimum of 7 tasks (2 activity associations, 1 track submission, and 4 track annotations) which were presented in a random order. In the submission and annotation subtasks, a random selection of 22 mood terms were included. Unlike in Experiment 1, the subset of terms included in the track annotation subtask was not the same as in the track submission subtask.

The elimination of affect term was performed based on two criteria: how frequently the terms were used in the track submission task (*submission frequency*), and how well the selected terms of the track submission subtask were agreed upon in the track annotation subtask (*recall rate*). The submission frequency was calculated as the number of subtasks where a term was selected divided by the number of subtasks where the term was included in the checklist. The recall rate was calculated as the number of tracks for which a term was selected in both the submission and annotation subtasks divided by the number of tracks for which the term was selected in the submission subtask. The rationale here was to create a frequency-independent value for each term, which reflects the communication accuracy of each term.

For EFA, the data from the annotation subtask was assembled into a matrix form where each row represented one response, and each column represented one affect term. Each row contained 22 binary values that reflected the terms which were randomly chosen and presented to the participant. The remaining 66 terms were represented by missing values to create a full array of the 88 terms described in Experiment 1. Missing values are generally accepted for EFA. Special consideration was given to the binary nature of the data, which is not suitable for typical Pearson correlation-based EFA [48]. Therefore, a polychoric correlation was used instead [49, 50]. EFA was run with the ordinary least squares (OLS) method to find the minimum residual, combined with the oblimin rotation. For CFA, response-level data was summarised into track level by computing for each mood term the percentage of positive answers, ignoring the missing values, and a diagonally weighted least squares (DWLS) method was used to test structural models on the data.

## Results

2508 participants submitted 5129 unique tracks (mean duration = 4 min 25s, 2456 unique artists, 3939 unique releases). Artists with the largest number of tracks were Madonna (35), Michael Jackson (35), and Eminem (34). The tracks submitted the most frequently were Pharrell Williams: Happy (from "Despicable Me 2") (81), P!nk:Just Give Me a Reason (68), and Shakira: Waka Waka (This Time for Africa) (45). On average, 211 activity associations were obtained for each mood term in the mood-activity association subtask (median count). From the submission subtask, each affect term was selected 89 times (median). In the annotation subtask, each affect term was selected 762 times (median). The affect term "happy" was again associated with activities the most frequently (37%), the term "in love" was the most frequently selected term for track submission (16% frequency), while "rhythmical" was again the most frequently applied term in the track annotation task (22%).

Of 88 mood terms, all terms that were infrequently chosen (<4% of the time) or not identified in the submission subtask (recall rate <.10) were discarded, resulting in 57 moods. Submission and recall thresholds were chosen on the basis of the elbow shapes in the curves of the respective sorted values. The factorability of the subtask-level data was inspected with Kaiser-Meyer-Olkin sampling measure adequacy (KMO), which yielded an excellent overall factorability (KMO = 0.80) [51]. However, two mood terms ("retro" and "epic", MSA 0.46 and 0.58, respectively) were observed to achieve values below the recommended measure of sampling adequacy (MSA, 0.60) [52], and therefore these terms were discarded. Models were constructed in R (version 4.3.1) (R Core Team, 2023) using lavaan (version 0.6.16) [53] and psych (version 2.3.9) [54] libraries.

Then, the polychoric correlation matrix of the remaining 55 mood terms was subjected to a parallel analysis [55] to determine the optimal number of factors. This produced an optimal of 21 factors and 17 components, accounting for 65% of variance (RMSEA = .529, TLI = -1.327). To explore the robustness of this initial structure, the CFA estimation was applied to track-

level data, appropriate for polychoric correlations [56]. A poor fit was obtained ($\chi^2$ = 3263.2, $p$ <.001, CFI = .839, RMSEA = .019, see the *Original* model in Table 2). A trimming of the model was carried out to improve the consistency and generalisability of the model. Based on response level data, the alphas of the items (affect term) for the factors were estimated and those items within the factors that brought the overall alpha below a suggested threshold (.60, [57]) were eliminated. Furthermore, if the elimination of elements within a factor did not increase the total alpha of the factor above the threshold, the factor was eliminated. This trimming resulted in the elimination of 16 items and three factors, and noticeable improvement in the model fit ($\chi^2$ = 1032.2, $p$ < .001, CFI = .905, RMSEA = .016, see *Alpha model* in Table 2).

We also outlined a model inspired by retaining the most common affect terms (43 in total) with 14 factors to maximally preserve the diversity and richness of the potential affect space. This model, named *Manual*, was the result of a manual pruning of factors in an attempt to retain the most frequently used terms in the analysis. The fit of the manual model was modest ($\chi^2$ = 1544.6, $p$ <.001, CFI = .910, RMSEA = .015), providing a better fit than the *Original* and *Alpha* models ($p$<.001, all model comparisons with $\chi^2$ tests).

Evaluating the 14 factor *Manual* model exposed the need for further clarity to make a more parsimonious model. Five factors which contained both positively and negatively loaded items (e.g., "soft" + "angry") were each split into two factors. Additionally, eight factors with only single items were combined into other factors by assessing the increase in the factor alpha value, culminating with a selected factor structure that yields the maximum increase (e.g., "soft" was combined with "free" + "easy" and "smooth"). This operation produced a combined factor model with a minor improvement in the model fit ($\chi^2$ = 993.8, $p$ < .001, CFI = .925, RMSEA = .013; see the model *Combined* in Table 2).

In the final attempt to streamline the *Combined* model, we looked at the contributions of the affect terms within the model in the CFA (standardised model parameters) and eliminated items below the 20% quantile value (0.21). This resulted in elimination of eight terms ("liked", "free", "easy", "rhythmical", "euphoric", "celebratory", "motivational", and "successful") and five factors with singular terms ("sexual", "sensual", "heroic", "spiritual" and "dramatic"). After this, we have an *Optimal* model with 7 factors and a total of 23 terms, which obtained a good fit to the data ($\chi^2$ = 409.1, $p$ <.001, CFI = .956, RMSEA = .014). The factors were labelled with one of the constituent affect terms that was the search term most frequently used within the factor (see Fig 3). The frequency of the terms and their retrieval accuracy in the track annotation task, as well as associations with the sets identified above (a set of 55 and 23 terms) are shown in S4 Table in S1 File.

**Table 2. Factoring fit indices in CFA at different stages: After the parallel analysis (*Original*), after alpha factoring (*Alpha*), after combining single item factors (*Combined*), and after removing low loading items (*Optimal*).** Three existing models (*MIREX*, *GEMS*, and *GEMIAC*) representing emotion structures induced by music are shown for comparison. $\chi^2$ is chi square difference statistic, Df degrees of freedom, the CFI comparative fit index, the TLI goodness of fit index, the RMSEA root mean square error of RMSEA with 90% the confidence intervals.

| Model | Factors | Items | TLI | $\chi^2$ | df | $p$ | CFI | RMSEA (90% CI) |
|---|---|---|---|---|---|---|---|---|
| Original | 21 | 55 | .805 | 3263 | 1224 | <.001 | .839 | .019 (.018-.019) |
| Alpha | 18 | 36 | .867 | 1032 | 449 | <.001 | .905 | .016 (.015-.018) |
| Manual | 14 | 43 | .894 | 1544 | 769 | <.001 | .910 | .015 (.013-.016) |
| Combined | 12 | 36 | .911 | 993 | 532 | <.001 | .925 | .013 (.012-.015) |
| Optimal | 7 | 23 | .946 | 409 | 209 | <.001 | .956 | .014 (.012-.016) |
| MIREX | 5 | 16 | .902 | 211 | 94 | <.001 | .924 | .016 (.013-.019) |
| GEMS | 9 | 22 | .894 | 423 | 173 | <.001 | .921 | .017 (.015-.020) |
| GEMIAC | 14 | 30 | .893 | 830 | 314 | <.001 | .923 | .019 (.017-.020) |

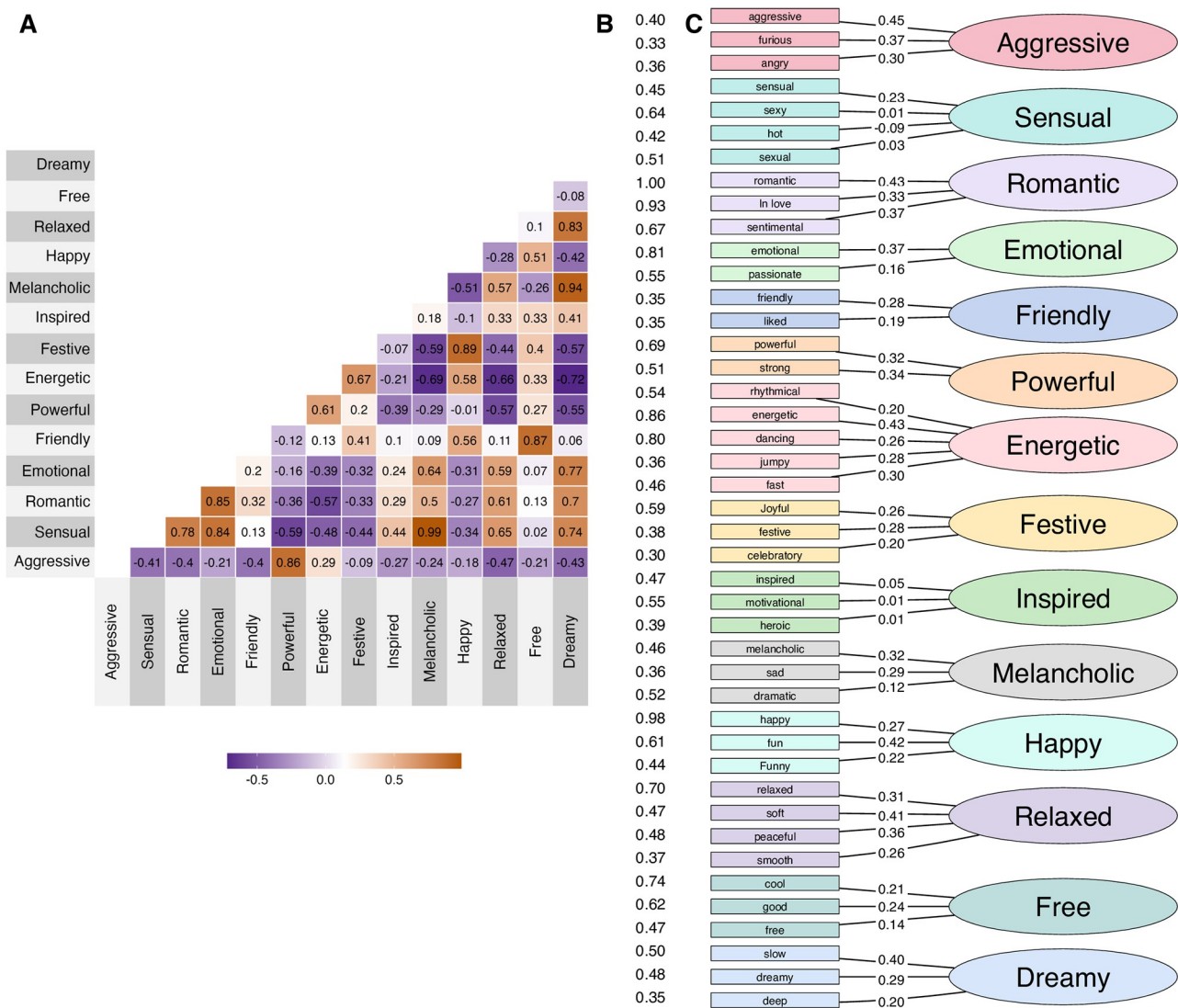

**Fig 3. Factor structure of the *Manual* model from CFA.** Panel A shows the correlations between the factors. Panel B displays the normalised search frequency of the affect terms, which has been used to label the factor (the name of the factor has been taken from the most frequent term). Panel C shows the standardised coefficients between the affect terms and the factors.

We examined whether the structures identified in previous research, mainly relating to induced emotions, were identified in the present materials (see S5 Table in S1 File) that correspond to nine dimensions of the *GEMS* model [3], five clusters of emotions in the so-called *MIREX* model [17], and fourteen dimensions of the *GEMIAC* model [34]. The remaining three models of induced emotions by music in the literature [4, 34, 35, 39] could not be implemented with the present affect terms because models specify several negative dimensions, which simply do not exist in the present structures (Shame-guilt, Anxiety-fear, and Disgust-contempt for Juslin's model, Indignant/defiant and Scary/fearful for Cowen's model, and Feeling of ugliness, Confusion, Uneasiness for Schindler's model). All of these models provided a satisfactory, but not good, fit with the data (*MIREX* CFI = .924, *GEMS* CFI = .921 and *GEMIAC* CFI = .923). Although these models aimed to deliver induced emotions by music

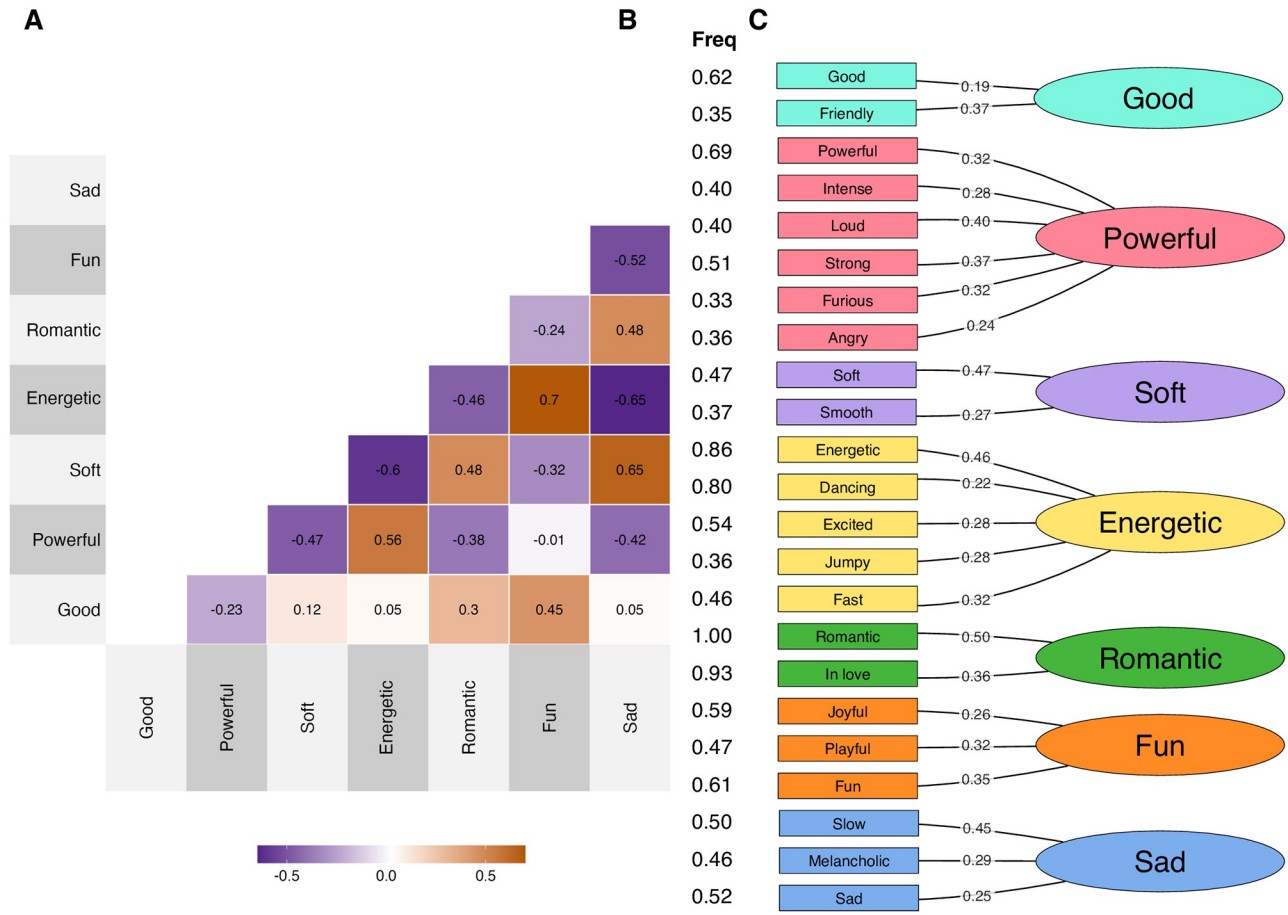

**Fig 4. Factor structure of the *Optimal* model from CFA.** Panel A shows the correlations between the factors. Panel B displays the normalised search frequency of the affect terms, which has been used to label the factor (the name of the factor has been taken from the most frequent term). Panel C shows the standardised coefficients between the affect terms and the factors.

were manually custom fitted with respect to affect terms and dimensions, none of them reached a good fit (CFI > .95) with the data, unlike the *Optimal* model.

The indicators for the CFA models in Table 2 show that while several models achieve an acceptable fit in terms of *Comparative Fit Index* (CFI > .90, [58] and RMSEA (<.05), only three models pass the *Tucker Lewis Index* (TLI > .90, [59] and only *Optimal* receives a robust CFI fit index (>.95, [60]).

Fig 3 displays the 14-factor (*Manual*) solution. This model captures affect structure with 14 dimensions, and although the fit of the model is acceptable (CFI > .90, RMSEA < .05), it may suffer from some poorly loaded factors (particularly inspired as a factor) and affect terms (e.g. "sexy" and "hot"). For comparison, Fig 4 displays the 7-factor (*Optimal*) solution that provided the best fit to the data in EFA. Factors have 2 to 7 affect terms within them, and factors' labels have been assigned according to term prevalence in the search task (shown in panel B, normalised according to the maximum number of searches, $S_{max}$ = 210 for romantic). Panel A shows the correlations between the factors. By contrasting these two solutions, it is clear that there is no simple mapping between the two solutions, as the underlying affect term co-occurrences lend themselves into distinct structures that may only partially overlap. In the General Discussion, we trace this linkage between the models (see Fig 6).

## Discussion

Consistent with the insights from Experiment 1, the majority of emotional expression is related to positive emotions (motivational, fun, romantic, energetic, soft, and good), and only perhaps two factors, Sad and Powerful, capture negative emotions expressed as melancholic, angry, or furious. This is not unlike the representation of induced emotions in the past studies; *GEMS* model [3] has 2 out of 9 dimensions as negative emotions (sadness and tension), while *AESTHEMOS* has 6 out of 21 factors related to negative emotions (Anger, Confusion, Boredom, Uneasiness, Sadness and Feeling of Ugliness, [4]). In the present structure, the *Good* factor demonstrates a departure from what the music-induced emotion models provide [3, 4, 35], nor has this factor explicitly been present in the affective circumplex model [41]. Terms such as "liked" and "good" may be construed as value judgements of the music, not as affect terms, but a participant could just as easily be referring to an affectual state of "feeling good", which has been observed in past studies [6]. There are some notable correlations between the factors, since our structure discovery method allowed for oblique construction of factors using oblimin rotation. Fun and energetic correlate positively ($r = .74$), powerful and energetic, and Good and Fun also show considerable correlations ($r = .57$ and .51, respectively). In addition, the factors labelled Sad and Soft, as well as Sad and Romantic, correlate positively ($r = .64$ and .49, respectively). This could be interpreted as showing that there is still potential to further reduce the dimensionality and explore whether some fundamental core affects, such as negative positive or high-low arousal, could also capture the expressed emotions represented in these data. Such an exploration would benefit from further data, which we detail in Experiment 3.

## Experiment 3

The aim of Experiment 3 was to validate the *Optimal*, 7-factor model obtained in Experiment 2. To achieve this objective, we collected data using the wider but more compatible *Manual* factor model, described in Experiment 2, with 14 factors to avoid misspecification of a factor structure [61]. Obtaining annotations of each factor in the track annotation subtask for the same tracks collected in Experiment 2. We also apply EFA and CFA to explore the possibility of a meta-factor model with only a few factors. And finally, we want to draw comparisons with existing frameworks (valence and arousal, and basic emotions) commonly used to describe emotions expressed by music.

## Method

**Materials.**  All tracks in Experiment 2 were used for validation after filtering out duplicate tracks based on track metadata (artist name and track title), resulting in 4780 tracks. In addition, 419 popular music excerpts (15s) from Saari and Eerola [32], called "set600" and 360 film soundtrack excerpts (30s) from Eerola and Vuoskoski [31] were included for external validation to be rated with the same terms as the rest of the materials.

Factor validation was achieved using CFA and correlations between factor annotations, and both individual item annotations, as well as inferred factor annotations from Experiment 2. To infer the factor annotations from Experiment 2 data, the mean across the factor elements was calculated.

**Participants.**  957 participants participated in the tasks, possessing demographics similar to those of Experiments 1 and 2; age M = 32.05, SD = 9.69, 71.2% with limited working proficiency in English or higher, and 75.6% were either non-musicians or music loving non-musicians. 69.9% of the participants identified as men and 28% from Asia, 43.2% from Europe, 7.1% from North America, and 20.2% from South America.

**Procedure.** Each task consisted of 10 track annotation subtasks where each of the factors was rated on a three-level Likert scale, asking "To what degree does the music example express the following moods?" (1 = "Not at all", 2 = "Somewhat", 3 = "A lot"), where the affects terms were presented as factor names followed by comma separated affect terms in parentheses, for instance, ROMANTIC (romantic, in love, sentimental). Data collection was carried out between 12/5/2015 and 30/6/2015. 48,596 ratings were collected, yielding a minimum of 10 ratings for each track.

## Results

First, we explore the degree to which the 14 factors in this experiment correlate with the affect terms that represent the same factors in Experiment 2 on all tracks ($N = 4780$). We also correlate the factors with the emotion concepts that are frequently collected in music emotion recognition experiments, including three affect dimensions (valence, arousal, and tension), which are available for *Set 600* ($N = 419$, [32]), and five basic emotions (anger, fear, happy, sad, and tender), which are available for *Film Soundtracks* ($N = 110$, [31]).

This analysis, shown in Table 3, suggests that the affect terms in Experiment 2 relate consistently positively, albeit moderately weakly, to the factors in the present experiment although, before comparing these to internal correlations within the experiment (factors and dimensions and basic emotions), the magnitude of these correlations is low due to the nature of the data in Experiment 2 where the aggregation of the presence of the affect term for each track has been compiled, and the amount of data is 10 to 40 times larger than with the comparison of ratings within the present experiment (the rest of the table). Factors such as Aggressive, Energetic and Romantic are particularly well matched between the terms in Experiment 2 and the present factors, while factors such as Friendly ($r = .09$, $p <.0001$), Free ($r = .08$, $p <.0001$) and Inspired ($r = .07$, $p <.0001$) bear a small similarity to Experiment 2.

**Table 3. Correlations between affect terms in Experiment 3 and emotion concepts from previous studies, where Exp. 2 refers to the same terms in the track annotation task in Experiment 2 of the present study ($N = 4780$), set 600 provides ratings of three affect dimensions ($N = 419$ [32]), and Soundtracks offer ratings of five discrete emotions ($N = 110$ [31]).**

| Factor | Exp. 2 | Dimensions (Set 600) | | | Discrete emotions (Soundtracks) | | | | |
|---|---|---|---|---|---|---|---|---|---|
| | Terms | Valence | Arousal | Tension | Anger | Fear | Happy | Sad | Tender |
| Aggressive | .44 | -.53 | .48 | .55 | .61 | .33 | -.10 | -.30 | -.29 |
| Dreamy | .28 | .15 | -.69 | -.54 | -.49 | -.32 | -.04 | .33 | .53 |
| Emotional | .18 | .15 | -.49 | -.46 | -.32 | -.32 | .19 | .10 | .38 |
| Energetic | .44 | -.09 | .75 | .51 | .27 | .07 | .36 | -.41 | -.34 |
| Festive | .22 | .28 | .44 | .07 | -.16 | -.29 | .58 | -.26 | -.08 |
| Free | .08 | .32 | .22 | -.12 | -.15 | -.27 | .40 | -.17 | .07 |
| Friendly | .09 | .40 | -.02 | -.32 | -.22 | -.25 | .39 | -.15 | .11 |
| Happy | .27 | .43 | .41 | -.03 | -.18 | -.27 | .56 | -.30 | -.04 |
| Inspired | .07 | .08 | -.16 | -.16 | -.14 | -.32 | .32 | -.04 | .14 |
| Melancholic | .23 | -.03 | -.69 | -.46 | -.17 | -.11 | -.22 | .35 | .34 |
| Powerful | .32 | -.35 | .55 | .52 | .54 | .18 | .03 | -.26 | -.35 |
| Relaxed | .34 | .25 | -.64 | -.58 | -.57 | -.50 | .12 | .37 | .62 |
| Romantic | .37 | .21 | -.60 | -.59 | -.39 | -.42 | .21 | .08 | .52 |
| Sensual | .16 | .16 | -.34 | -.35 | -.18 | -.26 | .11 | .05 | .29 |

*Note*. Statistical significance: Exp. 2 $|r| > .05$ $p<.001$, Dimensions $|r| > .17$ $p<.001$, and Discrete emotions $|r| > .31$ $p<.001$.

It should be noted that several of the external concepts seem to tap into the newly identified 14 terms, namely Arousal for Energetic ($r$ = .75) and ratings of happy as a basic emotion with a similarly named construct of Happy ($r$ = .56). Dreamy is similar to the lack of arousal ($r$ = -.69) and lack of tension ($r$ = -.54). Dimensional concepts tend to show higher correlation ($|\bar{r}| = .36$) with the 14 affect terms than with the basic emotions ($|\bar{r}| = .27$). There are also some terms which are poorly correlated with either of the conventional frameworks such as Inspired (all $|r| < .32$) and Sensual (all $|r| < .35$). However, these cross-dataset comparisons may reflect the selection criterion of the datasets, which have been constructed to provide good examples of the concepts of dimensions and basic emotions [31].

Moving on to structure discovery, the EFA procedure was similar to that in Experiment 2 and was carried out on the rating data (48596 observations for the 14 terms). The KMO measure of sampling adequacy was examined for subtask-level data, which shows excellent general factorability (KMO = .86). The measure for each item (factor) was above the suggested threshold of .60 and therefore no factors were discarded from the metafactor EFA. Parallel analysis of the polychoric correlation matrix suggested an optimal of 4 factors and 3 components, accounting for 59% of variance (RMSEA = .064, TLI = .943, $\chi^2$ = 8096).

Factor I captures Happy (.79 as factor score), Friendly (.57), Festive (.54), and Free (.46). Factor II consisted of Romantic (.85), Emotional (.64), Sensual (.62) and Melancholic (.43), Factor III Relaxed (.79) and Dreamy (.70), and Factor IV Powerful (.82), Aggressive (.76) and Energetic (.57). We will call this the *Meta* model, as it identified four factors with 13 elements, only failing to include Inspired in the factors. Factors in the *Meta* model loosely resemble 2 out of 4 quadrants from the affective circumplex space; the Factor I shows positive valence and high arousal type affect (quadrant 1) and Factor III shows positive valence and low arousal (quadrant 2). However, Factors II and IV do not quite fit the traditional affective circumplex quadrants 3 and 4, which are situated on the negatively valenced side of the space, probably due in part to some terms in these factors not being negatively valenced. Factor II has a low arousal and negative term, "Melancholic", which would be appropriate for quadrant 3 but the overall factor seems to cover more positively valenced expressions ("Sensual", "Emotional", and "Romantic"). Similarly, Factor IV contains one appropriately negative valenced and high arousal term, "Aggressive", that fits quadrant 4, but the rest of the terms suggest a positively valenced connotation, which does not align with a typical quadrant 4. It may be worth noting that a previous large-scale analysis of emotion tags of music [20] showed consistent positive bias of affect terms when projected into the affective circumplex, including terms such as "melancholic", to be positively valenced when music is concerned.

Another observation about the *Meta* factors is that they bear resemblance to the structure of five music preference factors (known as MUSIC) as outlined by Rentfrow, Gosling, and Levitin [62]; Factor I seems to resemble Contemporary music preference factor structure, Factors II and III are akin to combinations of Mellow (relaxing, smooth) and Unpretentious (romantic, sad, complicated) facets, and Factor IV bears close semblance to Intense music preference factor (loud, powerful, energetic music). It is possible that the contextualised way of collecting the affect term data has aligned these to be partially overlapping with the broad structure of music preferences, which themselves are grounded in uses of music.

In the final stage, we applied CFA to *Optimal*, *Meta* and three alternative expressed emotion models to the raw ratings in Experiment 3. Alternative models were created by assigning the constructs in the four affective quadrants (*Quadrant*) and two simpler models, one where the terms of the low and high arousal constructs were separated (*Arousal*) and another with the positive and negative constructs separated (*Valence*), all based on mean scores from Warriner and Brysbaert [47] across the terms. The results of the CFA (Table 4) indicate that the *Optimal* model with seven factors provides the best fit in all measures and is the only model that

**Table 4. Factoring fit indices in CFA for different models.** *Optimal* refers to 7 factor optimal model from Experiment 2, *Meta* is a reduction of the present dataset into four meta-factors, *Quadrant* is one where the 14 affect term clusters have been organised into four quadrants in affective circumplex based on the valence and arousal values of the individual terms, and *Arousal* and *Valence* are two factor models.

| Model | Factors | $\chi^2$ | df | *p* | CFI | RMSEA (90% CI) |
|---|---|---|---|---|---|---|
| Optimal | 7 | 1721 | 26 | <.001 | .984 | .037 (.035-.038) |
| Meta | 4 | 9979 | 59 | <.001 | .940 | .059 (.058-.060) |
| Quadrant | 4 | 20475 | 71 | <.001 | .897 | .077 (.076-.078) |
| Arousal | 2 | 11431 | 43 | <.001 | .894 | .074 (.073-.075) |
| Valence | 2 | 31469 | 43 | <.001 | .708 | .123 (.121-.124) |

exceeds the thresholds for a good fit (CFI > .95, RMSEA < .05). The *Meta* model with 4 factors obtains a satisfactory fit but is inferior to the *Optimal* model (direct comparison, $\chi^2$ = 1722, $p < .001$). The three alternative models that were based on the affective circumplex models (*Quadrants*, *Arousal*, and *Valence*) failed to capture the data (all CFI below .90), although the *Quadrant* model narrowly showed an acceptable fit (CFI = .894).

To illustrate the differences of the two best models (*Optimal* and *Meta*), Fig 5 shows the correlation between the 14 terms using track-level data with 4780 observations, where the organisation of the terms follows the *Meta* model and the *Optimal* models. It should be noted that the broad affect structure picked up by the four factors in the *Meta* model is clearly demarcated by the pattern of correlations, but this structure is inconsistent with the *Optimal* model in terms of how Factor II (Sensual, Emotional, Romantic, Melancholic) in the *Meta* model is divided between 3 factors in the *Optimal* model (E = Relaxed, F = Romantic, and G = Sad). There are also other divisions, such as Energetic being a separate factor in the *Optimal* model but belonging to Factor IV (Energetic, Powerful, and Aggressive) in the *Meta* model. These discrepancies suggest that models identified are not strictly speaking hierarchical, as they offer non-collapsable structures based on the level of which the model abstraction is done.

## Discussion

The present experiment collected a data set with 14 terms across all unique tracks in Experiment 2. A reduction to seven factors developed in Experiment 2 provided a good account of the data. Furthermore, a new simplification using EFA suggested that four factors could account for the 60% variance in the ratings, although these two structures were not fully compatible. The plausible alternative accounts based on affective circumplex quadrants did not provide sufficient solutions to the data.

## General discussion

The richness of emotions that music can express has been eclipsed by the interest in the emotions that music can induce in listeners. As these loci of emotions are different [5], it has been a sensible move to focus on the induction, as the emotional experiences of the listeners are surely the most absorbing question in the topic. However, this implicit decision to focus on emotion induction and not emotion expression has created a remarkable rift in the literature; A telling example is "love", which is one of the most frequent themes in pop music [63, 64] and has been shown to be the top emotion thought to be expressed by music [6, 7], but "love" does not appear in any of the models of induced emotions. It may, depending on your viewpoint, be present through related affect factors such as Tenderness or Nostalgia that are represented in several emotion induction models such as GEMS or GEMIAC. This is just a single

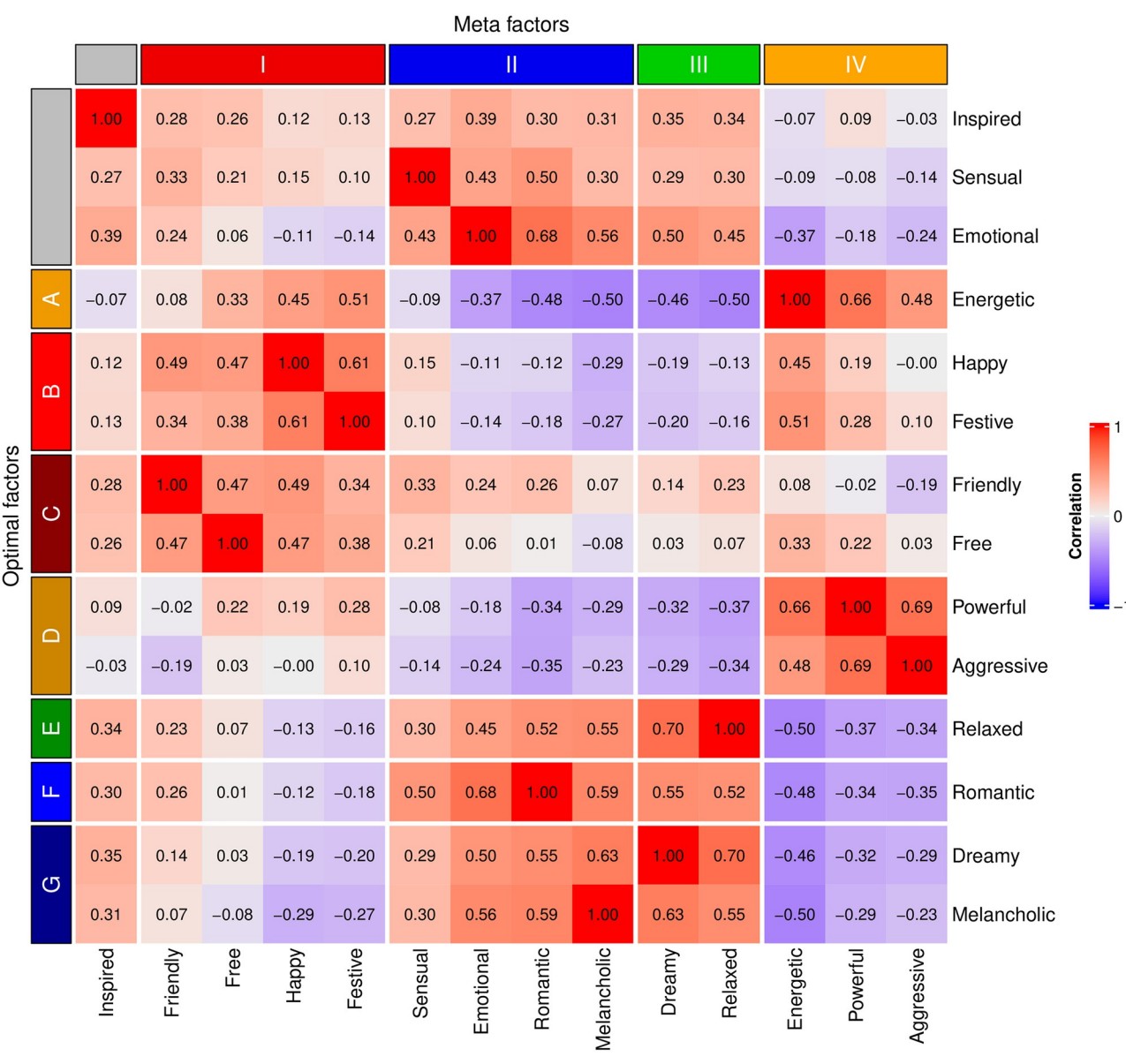

**Fig 5. Factor structure of the *Optimal* and *Meta* models on track-level ratings of 14 terms.** The heatmap shows correlations, and the annotation highlights both models.

example of how the terms and constructions of emotion induction and expression may not align.

In the present study, we have identified the appropriate affect terms that music may express and communicate by relying on a large collection of affect terms that were associated with music, activities, and actual music excerpts in an iterative paradigm carried out in three experiments. This bottom-up process allowed us to establish what affects were relevant for emotional expression. The resulting co-occurrences of the terms provided insight into the underlying structures of emotional expression in music. This process also attempted to contextualise the use of music by linking the terms with everyday activities. Although data and processes are different, especially when compared with past studies that have capitalised on Likert-type scales

to make it easier to establish factor structures, the advantage of the present paradigm allowed for questions to be kept grounded on music and its use. Through the subtasks, one could associate affect terms with actual music tracks and typical activities, and even allowed participants to identify the affects chosen by others to represent a musical track. To put this another way, the present approach emphasises what music affords to a listener [65], that arise through the dynamic interaction between listener and music itself. The expressive potential of music continues to be at the centre of musical affordances, promoting the idea that we capture expressions connected to actual situations, activities, and contexts [66, 67]. However, our approach did not cover the full range of activities that have been associated with music [2] such as meditation, sleep, or preparing for a fight were absent from our list of activities. Offering a larger palette of activities and creating realistic scenarios with active triggering of the activities would undoubtedly enrich the spectrum of relevant affect terms, perhaps widening the scope of the structures discovered or providing a granularity of affect within activities. We also wish to highlight that we did not exclude any music from our approach and allowed participants to suggest music, even if that music contained lyrics. In prior research, it has been common to exclude or avoid music with lyrics because the lyrical meaning could be considered an extra-musical feature, beyond the immediate interest of a study involving both emotion induction and emotional expression through music [68, 69].

To summarise our findings on this multilevel *Structure of Expressed Emotions in Music* (SEEM), Fig 6 visually links the levels of descriptions and analyses in the 3 experiments. At level A, we identified approximately 200 terms that are consistently used to refer to emotions in music, which form 88 clusters of terms. At level B, we establish that these 88 clusters of affect terms can be largely (65%) explained by 21 factors. Further optimisation (levels C and D) of the 43 affect terms led to the development of a 14 factor model and a reduction structure into 7 factors. Finally, an additional reduction in four factors (level E) was explored that accounted for 60% of the variance in Experiment 3. We do not suggest that this is a radically overhaul of the affect structures studied in this field, but at the same time we draw attention to the fact that many of the terms frequently chosen by participants such as "in love" and "romantic" are not well represented by the existing emotion models used to describe expressed emotions in music. We also wish to remain uncommitted to specific level of description of affects; having a structural description that operates on multiple levels (from 213 relevant terms to 55 consistent terms to 21, 14, 7 or 4 factors capturing these) is a beneficial property of representation that has to serve multiple purposes (annotation, tagging, and rating tasks to discriminate and identify music expressive different qualities).

We acknowledge that this bottom-up process will generate affect structures that reflect the sample, their cultural background, musical choices, and genre preferences. Although this sample is not perfect, at least the sample is a diverse, large predominantly Western sample that does not particularly rely on music experts or participants solely from Europe or North America. A similar iterative process would be informative to undertake with samples obtained in specific regions and cultures. The discovery of structures in this work could have been done differently, which may have led to minor variants of the structures. It would be interesting to conduct a structure discovery that is entirely based on hierarchical clustering [70, 71] to keep the layers fully collapsable or to extract orthogonal components in the first place [35]. Due to the nature of the data and initial exploration with such alternatives, we felt that the current iterative paradigm with EFA and CFA steps that incorporate trimming the affect term space operated logically and effectively, but we welcome other formulations on the data, which is freely available (https://dx.doi.org/10.5255/UKDA-SN-852024, see [72]) with full documentation.

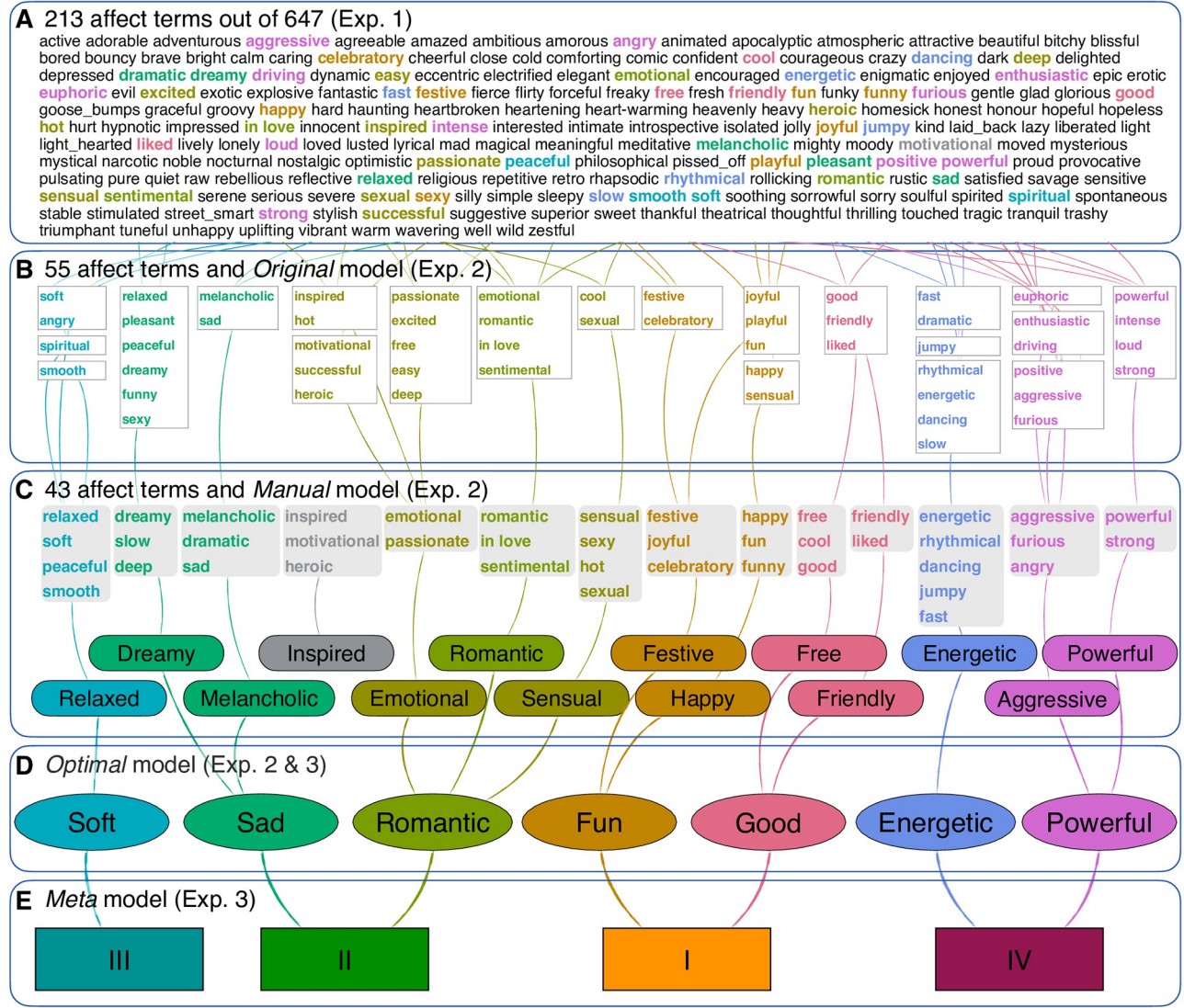

**Fig 6. Summary of the mapping of the terms into different structures across the experiments.** A shows the relevant affects (213 out of 647), B displays the initial 21 factors with associated affects (*Original* model), C shows a reduction into 14 factors (*Manual* model), D into 7 factors (*Optimal* model), and E into 4 factors (*Meta* model).

The present study has offered a set of alternative structures and affect constructs to explore what emotions music can express in the context of Western music. This multilevel structure of expressed emotions in music allows us to focus on concepts and terms that are usually over-looked, such as "in love", "festive", or "friendly", which form an integral part of the optimal structures that are seldom identified in research involving expressed emotions [73–76], although some of these have been recognised in past research [6]. However, the newly identified structures also reproduce several familiar and conventional concepts that resemble basic emotions or affective circumplex concepts such as "sad", "energetic", and "powerful", and we demonstrate how these can be largely recovered from valence and arousal ratings. It is also worth acknowledging that structures identified in the domain of music-induced emotions [3, 4, 34, 35] clearly provide a disparate set of emotion dimensions or clusters, which typically

have distinct negative factors and a limited set of descriptive terms within them. The predominantly positive affectual terms and affect structure that our finding shows could be the result of either our minimal contextualization of a limited number of activities in the activity association task or because we allowed people to freely choose their appropriate affect terms across all stages of the process. These design choices may have predisposed participants to choose positively loaded terms. The bias of positivity itself is not uncommon in reported music and emotion studies [7, 20, 77], although we must acknowledge that another method with different contextual triggers would have generated a different set of results. For example, the situational function of music could just as easily have been made to express emotions of fear (as in horror films) or heartbreak (as in a case of lost love).

A related issue to discuss is whether the methodological choices used in establishing both music-induced and music-expressed emtions are sufficient to generate a divergence in structures. Several reanalyses of the studies that contained emotions felt and expressed by Schubert [78] suggested that although emotions largely match, the intensity of experienced emotions tends to be lower than the expressed emotions. He uses dissociation theory to explain why the two can sometimes diverge [78], especially in emotions such as sadness or fear. Furthermore, felt emotions are known to be more influenced by context and situational signals [79], as well as cognitive appraisal [80, 81].

Being able to describe the emotion expressed by music, in a nuanced way, has a host of applications ranging from everyday contexts of music use (e.g., matching music a desired or relevant expression of emotion) to health and well-being applications. The later of which may benefit from the present SEEM framework because of the increased resolution of the positive emotional expressions captured. Much of the Music Information Retrieval research on music emotion recognition would benefit from updating the recognition goals to create more plausible playlists and mood recommendation algorithms [82, 83].

It may be that the structure of the expressed emotions discovered is just one focused on consumers of music in western countries at this moment in time. However, we think that the iterative process we outlined could be fruitfully applied to other cultural contexts; the process does not impose the emotion framework nor the specific terms or dimensions on participants and keeps the everyday uses and the actual music within the diagnostic process. Additional research to contextualize the use of music would also be fruitful for broadening our understanding of how the same piece of music may express emotion differently, depending on the context. This offers the tantalising prospect of discovering how people in other cultures or subcultures, geographical regions, languages, or other relevant boundaries organise, perceive, and communicate emotions expressed by music.

## Supporting information

**S1 File.**
(PDF)

## Author Contributions

**Conceptualization:** Tuomas Eerola, Pasi Saari.

**Data curation:** Tuomas Eerola, Pasi Saari.

**Formal analysis:** Tuomas Eerola, Pasi Saari.

**Investigation:** Tuomas Eerola.

**Methodology:** Tuomas Eerola, Pasi Saari.

**Project administration:** Tuomas Eerola.

**Software:** Pasi Saari.

**Supervision:** Tuomas Eerola.

**Writing – original draft:** Tuomas Eerola.

**Writing – review & editing:** Tuomas Eerola, Pasi Saari.

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
