## [Decision Letter · Decision Letter 0]

3 Jun 2024

PONE-D-24-10867What Emotions Does Music Express? Structure of Affect Terms in Music Using Iterative Crowdsourcing ParadigmPLOS ONE

Dear Dr. Eerola,

Thank you for submitting your manuscript to PLOS ONE. After careful consideration, we feel that it has merit but does not fully meet PLOS ONE’s publication criteria as it currently stands. Therefore, we invite you to submit a revised version of the manuscript that addresses the points raised during the review process.

Two experts in the field have carefully reviewed the manuscript entitled “What Emotions Does Music Express? Structure of Affect Terms in Music Using Iterative Crowdsourcing Paradigm”. Both the reviewers and I found the manuscript very interesting and relevant for music emotion research. However, one of the reviewers has made observations that need to be addressed.

In light of these reviews and my own reading of the manuscript, I am requesting a major revision and resubmission, in which you will need to respond to each point made in the reviews.

Let me focus on some points that the reviewers and I would like to see addressed. These are:

1) The rationale for the methodology, in particular the motivation, strengths and weaknesses of selecting expressed emotions by their association with types of music-related activities. 

2) Improving the quality of the writing in English. 

3) Some minor points from me: 

Line 328, “5129 unique tracks that added up to 5129”, is it correct?

Experiment 3, line 445: you say that the aim is to validate the Optimal model, but you use the factors from the Manual model instead. This is confusing for the reader. 

Lines 519-522: the association with the four Valence-Arousal quadrants seems problematic, specially association Factors II and IV with negative valence emotions. 

There are other points brought out in one of the reviews and I will carefully attend to your item-by-item responses to them.

 Please submit your revised manuscript by Jul 18 2024 11:59PM. If you will need more time than this to complete your revisions, please reply to this message or contact the journal office at plosone@plos.org. Please include the following items when submitting your revised manuscript:A rebuttal letter that responds to each point raised by the academic editor and reviewer(s). You should upload this letter as a separate file labeled 'Response to Reviewers'.A marked-up copy of your manuscript that highlights changes made to the original version. You should upload this as a separate file labeled 'Revised Manuscript with Track Changes'.An unmarked version of your revised paper without tracked changes. You should upload this as a separate file labeled 'Manuscript'.

We look forward to receiving your revised manuscript.

Kind regards,

Bruno Alejandro Mesz, Ph.D.

Academic Editor

PLOS ONE

Journal Requirements:

P.S. was by funding from ESRC (ES/K00753X/1). 

4. Thank you for uploading your study's underlying data set. Unfortunately, the repository you have noted in your Data Availability statement does not qualify as an acceptable data repository according to PLOS's standards.

5. Please respond by return e-mail with an updated version of your manuscript to amend either the abstract on the online submission form or the abstract in the manuscript so that they are identical. We can make any changes on your behalf.

7. We notice that your supplementary figure 1 and tables 1 - 5 are included in the manuscript file. Please remove them and upload them with the file type 'Supporting Information'. Please ensure that each Supporting Information file has a legend listed in the manuscript after the references list.

Reviewers' comments:

Reviewer's Responses to Questions

**Comments to the Author**

1. Is the manuscript technically sound, and do the data support the conclusions?

Reviewer #1: Yes

Reviewer #2: Yes

2. Has the statistical analysis been performed appropriately and rigorously? 

Reviewer #1: Yes

Reviewer #2: I Don't Know

3. Have the authors made all data underlying the findings in their manuscript fully available?

Reviewer #1: Yes

Reviewer #2: Yes

4. Is the manuscript presented in an intelligible fashion and written in standard English?

Reviewer #1: Yes

Reviewer #2: No

5. Review Comments to the Author

**Reviewer #1:** The paper presents an original research about the emotions expressed by music, an area underexplored in the literature of psychology of music. The main contribution of this paper is the design of an iterative method that aims at overcoming previous setbacks on experimental research in the field. The paper is clearly and consistently written. Detailed information about arguments and hypotheses, method, and data analyses is provided. Conclusions and implications are fully supported by the results. The experiments were conducted rigorously. Criteria about all the decisions made along both the experimental phase and the analyses of results were explained in detail. Supporting information is provided and protocols and materials are also available for replication in public repositories.

**Reviewer #2:** This is a very interesting and rigorously presented research study on emotions associated with music. The statistical analyses are well documented and seem to be done properly, but I am no expert so I cannot confirm this with full confidence.

My main concerns and request for revision are the following:

Abstract: you state that the structure is not well aligned with existing models, so what is does not relate to, but you do not say what it consists of instead. This does not yet provide a positive and clear statement of the novel contribution.

1) The methodology employed in the research is quite different from previous work that aimed at finding the main emotion terms that can be associated with music. It is not clear how deliberate this is. Are the authors arguing that this way of identifying emotion terms is more valid than previous surveys? If so, this should be part of the discussion of previous approaches in the literature review and introduction. Or was the study originally conceived to serve a different purpose (more about the emotional functions of music in everyday life) and was (part of) the data collected used for the purpose of identifying expressed emotions in music? If so, this should be acknowledged more openly.

2) Related to point 1, I feel there should be a more transparent discussion of the strengths and weaknesses of the approach taken in this study in the discussion. The way these terms were chosen, are they strictly speaking about what emotions music can express or are they indicating the affects and emotions that music can afford to listeners in certain circumstances? Such a relational perspective can be seen as a strength of the approach. It is about what music affords to listeners rather than purely about the potential communicative power of music and its characteristics. The role of lyrics in this sample of music should also be acknowledged.

3) More generally what are the limitations that the approach taken may have promoted? The terms are chosen first of all in relation to activities that they can be associated with, and only quite a small number of activities are included. For example, mediation, sleep or connecting spiritually are not included, nor are having sex or preparing for war. A larger range of activities could have shown a larger range of expressions?

4) The literature review is not exhaustive. Whilst that may not be the purpose of the literature review in this case, the writing does suggest as if all relevant most recent literature has been included. However that is not the case – the below is for example also a relevant article. Also, did not GEMS also include expressed emotion? The 11 terms used in Laukka et al. cross-cultural study (2013, Emotion) seems also relevant?

Schubert, E. (2003). Update of the Hevner adjective checklist. Perceptual and motor skills, 96(3_suppl), 1117-1122.

5) A discussion of the terms is important too – affect terms, emotions, or even more broadly? The introduction is formulated in a way that suggests that this research will give an updated and more definitive list of emotions terms expressed by music. These promises should be checked with what the study actually offers in the end including in light of its limitations.

6) Abstract also needs to be updated in light of these comments.

5) The English is not yet good enough. There are a lot of awkward sentences with poor grammar, which makes the reading challenging and reduces the clarity of what is written.

Minor comments:

P2/30 l42. ‘inferior model’. This seems a bit strongly formulated. Also other models? How many, which do you mean?

P3/30 l71. Some missing references such as to Hevner and Schubert.

Figure 1 legend – Level a refers to the input ‘of’ or input ‘to’? I think it should be ‘to’.

L127/8. Rather abstract first sentence under Rationale. Please improve.

L141. You regularly state ‘aim’, which makes it confusing what the aim of the research is.

L150. Why is it important to collect this contextual information. This is buried in the methodology and needs more explicit discussion (see general comments).

L162. Limited working proficiency? What is meant by that? Are you talking about English language proficiency?

L169. Western music and classical music -> clarify this further, be more precise.

L171. You present various statistics here but have not yet explained that you would be collecting those. Add a signposting explanation before summarising the data.

L225. Low ability to associate tracks – could it be that some search terms are less successful in retrieving an appropriate track, and could this be a confounding factor?

L243 Silhouette scores? More info is needed.

L247 not clear what analysis was done to reduce to 88 clusters. There were 9 activities, so how did those play a role in the reduction as suggested above?

Figure 2. Panel A, B and C are not matched in order with the order presented in the figure caption.

L264-5. Check grammar

L282 and elsewhere. Check what tense you use where. In Exp 2, method, you used past tense. Here you use present. (aim was / aim is).

L302. do you mean that participants did two times an activity association task and four times an annotation subtask? They could not do as many tasks anymore as they wanted? The way this is phrased is unclear as it can also be two different activity association tasks.

L307 and the rest of that section. This is data analysis, not data collection procedure.

L314. This procedure (division) is that to get a proportion? Can be further explained.

L322. Why binary terms and why 22. I am lost a bit here without further explanation.

L330. Not majority of the track, but the largest number of tracks?

L333. The median of 211 activities associations were obtained. This is phrased in a rather confusing manner. Please rephrase and clarify. Also, 211 unique associations or are just reporting how many responses? I assume the latter. But that is not so relevant, is it?

L373. ‘separated into two’. Do you mean each was split into to two to yield 10, or combined into two factors?

L374. by considering computing? Or do you mean by computing? Surely you do not just consider it?

L375. were attempted to combine is not good grammar

L421. In General Discussion. ‘the’ is missing.

L433. Did participants clearly chose music to express this? I would discuss and unpick this a bit further (see general comments).

L443. Further data – as will be done next.

L445-7. First sentence of the section is hard to parse. Please improve.

L479-480. check sentence the factors are collected in this experiment, are not they? Or perhaps 'with' is not the correct word?

Table 3. table heading. not clear what is being correlated as you say 2x 'between' in this sentence.

Table 3 and elsewhere: check when you do and do not include a 0 before the full stop. You say p < .01 but also r = 0.01. All numbers that range between 0 and 1 should be treated the same, so this would need to be r = .01 to be consistent. (and making the r and p italic).

L497. Dreamy is ‘simply’. Why simply and how can you be so sure?

L499. what do you mean by 'the more dimensional concepts?

L502. and inspired and sensual are themselves the most poorly defined?

L528 and elsewhere. affect terms or concepts? You switched terminology. Be consistent for clarity. Or is this change on purpose?

L534. acknowledge that quadrant is close to .9

L605. different sets of structures -> is this undermining the confidence in the outcomes too much? Formulate less strongly?

L625. ‘less descriptive’ should this be ‘fewer descriptive’?

S1 Table 1. Why this order of continents? It is not alphabetical nor in order of number of participants.

S1 Table 4. How can the rank be alphabetical?

6. PLOS authors have the option to publish the peer review history of their article (what does this mean?). If published, this will include your full peer review and any attached files.

Reviewer #1: **Yes: **ISABEL CECILIA MARTINEZ

Reviewer #2: **Yes: **Renee Timmers

---

## [Author Response · Author response to Decision Letter 0]

20 Jul 2024

Response letter

Two experts in the field have carefully reviewed the manuscript entitled “What Emotions Does Music Express? Structure of Affect Terms in Music Using Iterative Crowdsourcing Paradigm”. Both the reviewers and I found the manuscript very interesting and relevant for music emotion research. However, one of the reviewers has made observations that need to be addressed.

In light of these reviews and my own reading of the manuscript, I am requesting a major revision and resubmission, in which you will need to respond to each point made in the reviews.

Let me focus on some points that the reviewers and I would like to see addressed. These are:

1) The rationale for the methodology, in particular the motivation, strengths and weaknesses of selecting expressed emotions by their association with types of music-related activities. 

R: Thanks for the suggestion. We agree that we did not give sufficient background and justification to the task of nominating the expressed emotions and associating them with the music-related activities. We now have covered this in a better way in several sections of the manuscript, the introduction (p. 2), under “iterative paradigm” (p. 3) and the general discussion (p. 16). More specifically, we acknowledge that associating the terms with activities attempts to broaden the context of what kind of emotions music might be expressing as different activities require different emotional characters (preparing for an exercise has different expressive demands than trying to sleep or relax). We align these ideas with functions of music (Sloboda et al., 2009) and everyday uses of music (e.g., Randall & Rickard, 2017; Juslin et al, 2008).

2) Improving the quality of the writing in English. 

R: Fair point and we recruited the services of a professional proof-reader to go over the text, leading to numerous language improvements.

3) Some minor points from me: 

Line 328, “5129 unique tracks that added up to 5129”, is it correct?

R: This was poorly articulated although correct as participants in Experiment 2 submitted 5,129 unique tracks that exceeded the unique tracks (4,472) submitted by the participants in Experiment 1. But we agree that this reads awkwardly so we have altered the sentence to be clear.

Experiment 3, line 445: you say that the aim is to validate the Optimal model, but you use the factors from the Manual model instead. This is confusing for the reader. 

R: This is a great point and we clearly have not explained our rationale sufficiently clearly to the reader. Our aim is to validate the Optimal model but we collect data on a broader set of terms (i.e., Manual model) as it is better to have a larger set of items to avoid misspecification of the factor structure (Brown, 2015) and most EFA and CFA cycles recommend starting broad rather than narrow (Fabrigar, Wegener, MacCallum, & Strahan, 1999, p. 278) as underfactoring (having fewer factors) is much more damaging for the CFA than overfactoring (starting with additional factors) (MacCallum, Widaman, Zhang & Hong, 1999, p. 96). We have now clarified our rationale of collecting a wider set of items (Manual model) to assess the fit of the Optimal model in the beginning of the Experiment 3.

References

Brown, T. A. (2015). Confirmatory factor analysis for applied research. Guilford publications. 

Fabrigar, L. R., Wegener, D. T., MacCallum, R. C., & Strahan, E. J. (1999). Evaluating the use of exploratory factor analysis in psychological research. Psychological methods, 4(3), 272-299. 

MacCallum, R. C., Widaman, K. F., Zhang, S., & Hong, S. (1999). Sample size in factor analysis.. Psychological Methods, 4(1), 84-99. 

Lines 519-522: the association with the four Valence-Arousal quadrants seems problematic, specially association Factors II and IV with negative valence emotions. 

R: That’s a great point and we were trying to be careful of not equating the Meta model with four quadrants as we only mentioned that the factors “...loosely resemble the four quadrants”. But you are completely right, it was stretching the similarity too much and we have dismantled and downscaled this association by characterising it more accurately (...”loosely resemble 2 out of 4 quadrants…”) and now describe the similarities (Factor I = Q1 and Factor III = Q3) and expose and discuss the discrepancies more carefully.

There are other points brought out in one of the reviews and I will carefully attend to your item-by-item responses to them.

Reviewers' comments:

Reviewer's Responses to Questions

Comments to the Author

1. Is the manuscript technically sound, and do the data support the conclusions?

Reviewer #1: Yes

Reviewer #2: Yes

2. Has the statistical analysis been performed appropriately and rigorously? 

Reviewer #1: Yes

Reviewer #2: I Don't Know

3. Have the authors made all data underlying the findings in their manuscript fully available?

Reviewer #1: Yes

Reviewer #2: Yes

4. Is the manuscript presented in an intelligible fashion and written in standard English?

Reviewer #1: Yes

Reviewer #2: No

5. Review Comments to the Author

Reviewer #1: The paper presents an original research about the emotions expressed by music, an area underexplored in the literature of psychology of music. The main contribution of this paper is the design of an iterative method that aims at overcoming previous setbacks on experimental research in the field. The paper is clearly and consistently written. Detailed information about arguments and hypotheses, method, and data analyses is provided. Conclusions and implications are fully supported by the results. The experiments were conducted rigorously. Criteria about all the decisions made along both the experimental phase and the analyses of results were explained in detail. Supporting information is provided and protocols and materials are also available for replication in public repositories.

R: Thanks for the positive and reaffirming comments.

Reviewer #2: This is a very interesting and rigorously presented research study on emotions associated with music. The statistical analyses are well documented and seem to be done properly, but I am no expert so I cannot confirm this with full confidence.

My main concerns and request for revision are the following:

Abstract: you state that the structure is not well aligned with existing models, so what is does not relate to, but you do not say what it consists of instead. This does not yet provide a positive and clear statement of the novel contribution.

R: Thanks for highlighting this. We have revised the abstract to articulate more clearly the positives and how the structure of expressed emotions offers new terms and choices of structures that have not previously been available (from highly reduced to more granular and including concepts (“romantic”, “free”) and terms that have featured rarely in the literature (e.g. “dreamy”, “festive” and “in love”).

1) The methodology employed in the research is quite different from previous work that aimed at finding the main emotion terms that can be associated with music. It is not clear how deliberate this is. Are the authors arguing that this way of identifying emotion terms is more valid than previous surveys? If so, this should be part of the discussion of previous approaches in the literature review and introduction. Or was the study originally conceived to serve a different purpose (more about the emotional functions of music in everyday life) and was (part of) the data collected used for the purpose of identifying expressed emotions in music? If so, this should be acknowledged more openly.

R: Good point, and while the data collection clearly serves multiple purposes (what are the expressive terms but also for what are these terms used for in musical context referring to activities), the wider motivation was to be able to link participants choices of terms and tracks and check whether the terms actually communicate the intended emotions, and the possibility of connecting actual musical examples to the emotions expressed by them. In the past, the methods have focussed on the terms (what are the terms music can express in Juslin and Laukka, or in music induced emotion, how relevant are a long list of emotional terms for a sample of participants). We argue that these past efforts have limited utility because (1) they don’t actively widen the range of expressions/emotions considered which we do with activities/uses, and (2) they are regularly done without a reference to actual music, and (3) the choice of the most usable terms from a large possibility of term vocabulary can be more effectively refined when the terms are dynamically tested by their ability to communicate the intended expression across people. We did not want to antagonise the past research that in our view has collected emotion term data without any contextualisation and music (Zentner et al., 2008; Schindler et al., 2017; Juslin and Laukka, 2003) even though these studies have recognised these issues years after the studies and structures have been completed (Juslin et al., 2022; Strauss et al., 2024). We have now more explicitly outlined the multiple purposes of the dataset created and justified the role of functions as a useful way of covering the large and multifaceted expressive space of music.

2) Related to point 1, I feel there should be a more transparent discussion of the strengths and weaknesses of the approach taken in this study in the discussion. The way these terms were chosen, are they strictly speaking about what emotions music can express or are they indicating the affects and emotions that music can afford to listeners in certain circumstances? Such a relational perspective can be seen as a strength of the approach. It is about what music affords to listeners rather than purely about the potential communicative power of music and its characteristics. The role of lyrics in this sample of music should also be acknowledged.

R: This is a great suggestion, thank you. We appreciate the encouragement to discuss the strengths and weaknesses of the chosen approach more fully. In terms of the emotions expressed purely without the context and the emotions expressed in situations, context and activities, we think there is a limited value in pure expressive intention as most music is listened to and designed to be relevant for specific functions and contexts, although we could have gone much further in this by creating multiple scenarios where these terms might be relevant if we had attempted to fully embrace the context-sensitive of expressive purposes. But our approach is at least a minimal way of broadening the scope of expression, and even if we don’t fully situate and contextualise every decision to a real context, we nevertheless broaden the notion of what music might express by reminding the participant that there are a number of common activities done concurrently with music. In our view this is already better than asking participants in a single context (e.g., a classroom for Exp. 1-2 or in a music festival for Exp. 3 or at home for Exp. 4 in Zentner et al., 2008) or without a context (Juslin & Laukka, 2003). Thank you for reminding us about the importance of lyrics, which is often absent from music and emotion studies. We agree that spelling out the relational perspective is a useful way of articulating the purpose of the design and we have now included this in the discussion (and we do refer to some of the enactive work on this and acknowledge the lyrics).

3) More generally what are the limitations that the approach taken may have promoted? The terms are chosen first of all in relation to activities that they can be associated with, and only quite a small number of activities are included. For example, mediation, sleep or connecting spiritually are not included, nor are having sex or preparing for war. A larger range of activities could have shown a larger range of expressions?

R: Thanks for raising this issue and we agree that this was now a minimal contextualisation, and more could have been done to expand the range of expressions. However, wider contextualisation might best be done properly with different scenarios to invoke the many situations and contexts appropriately. As we cannot rerun the experiment, the issue of context is now something that we have mentioned in the General Discussion, and we acknowledge that the context was limited and a broader set of activities would probably have enriched the vocabulary and widened the scope of the emerging structures. It would be worthwhile to explore the impact of the context more fully in future empirical work.

4) The literature review is not exhaustive. Whilst that may not be the purpose of the literature review in this case, the writing does suggest as if all relevant most recent literature has been included. However that is not the case – the below is for example also a relevant article. Also, did not GEMS also include expressed emotion? The 11 terms used in Laukka et al. cross-cultural study (2013, Emotion) seems also relevant?

Schubert, E. (2003). Update of the Hevner adjective checklist. Perceptual and motor skills, 96(3_suppl), 1117-1122.

R: Thanks for these pointers and you are right, Schubert’s study (2003) needs to be – and has now been – acknowledged but the study by Laukka et al., (2013) is a test of 11 eclectically chosen emotional expressions (some basic emotions and some aesthetic emotions) that does not really answer the same questions (what terms, what structures). Zentner’s study (2008) did include emotional expression as part of the Experiment 2 but the paper only reports differences between felt and perceived emotions across genres, and unfortunately not the kind of outcomes that would have enabled us or others to deduce which expressions are relevant or what is the likely structure within them. 

This has been added to the introduction: “... An update to the taxonomies of emotion terms used since the 1930’s [21] and 1950’s [22] was proposed that consisted of 46 terms [23]”. And few lines later: “Since the two surveys conducted in Sweden [6,7], the update by Schubert [23] of the adjectives describing music defined by Hevner [21] and Farnsworth [22], … ”

5) A discussion of the terms is important too – affect terms, emotions, or even more broadly? The introduction is formulated in a way that suggests that this research will give an updated and more definitive list of emotions terms expressed by music. These promises should be checked with what the study actually offers in the end including in light of its limitations.

R: Good point and we on purpose keep the focus on affect terms and shy away from broader discussions of emotions as this would be more appropriate

---

## [Decision Letter · Decision Letter 1]

29 Sep 2024

PONE-D-24-10867R1What Emotions Does Music Express? Structure of Affect Terms in Music Using Iterative Crowdsourcing ParadigmPLOS ONE

Dear Dr. Eerola,

Thank you for submitting your manuscript to PLOS ONE. After careful consideration, we feel that it has merit but does not fully meet PLOS ONE’s publication criteria as it currently stands. Therefore, we invite you to submit a revised version of the manuscript that addresses the points raised during the review process.

Thank you for submitting your revision of your article “What Emotions Does Music Express? Structure of Affect Terms in Music Using Iterative Crowdsourcing Paradigm” I have read your revised manuscript, with the last round of reviewers’

comments in hand.  Thanks for you careful attention to their previous comments and my own. I am requesting you to address the point raised by the reviewer about expressed vs, felt emotion.

We look forward to receiving your revised manuscript.

Kind regards,

Bruno Alejandro Mesz, Ph.D.

Academic Editor

PLOS ONE

Journal Requirements:

Reviewers' comments:

Reviewer's Responses to Questions

**Comments to the Author**

1. If the authors have adequately addressed your comments raised in a previous round of review and you feel that this manuscript is now acceptable for publication, you may indicate that here to bypass the “Comments to the Author” section, enter your conflict of interest statement in the “Confidential to Editor” section, and submit your "Accept" recommendation.

Reviewer #2: All comments have been addressed

2. Is the manuscript technically sound, and do the data support the conclusions?

Reviewer #2: Yes

3. Has the statistical analysis been performed appropriately and rigorously? 

Reviewer #2: Yes

4. Have the authors made all data underlying the findings in their manuscript fully available?

Reviewer #2: Yes

5. Is the manuscript presented in an intelligible fashion and written in standard English?

Reviewer #2: Yes

6. Review Comments to the Author

Reviewer #2: My only remaining request is to improve the explanation of the rationale for the study in the abstract. Is it really the case that we know more about the structure of felt emotion than expressed emotion and has that really been more investigated? Such comparisons are hard to make. I would instead explain that expressed emotion in music is typically investigated without looking at the contexts in which music is experienced, which may lead to the uncovering of reduced and abstracted beliefs about expressed emotion rather than closely represented experiences of emotions expressed (also should it be affect instead?).

7. PLOS authors have the option to publish the peer review history of their article (what does this mean?). If published, this will include your full peer review and any attached files.

Reviewer #2: **Yes: **Renee Timmers

---

## [Author Response · Author response to Decision Letter 1]

16 Oct 2024

Reviewers' comments:

Reviewer #2: My only remaining request is to improve the explanation of the rationale for the study in the abstract. Is it really the case that we know more about the structure of felt emotion than expressed emotion and has that really been more investigated? Such comparisons are hard to make. I would instead explain that expressed emotion in music is typically investigated without looking at the contexts in which music is experienced, which may lead to the uncovering of reduced and abstracted beliefs about expressed emotion rather than closely represented experiences of emotions expressed (also should it be affect instead?).

R: Thanks for this helpful suggestion that clarifies the rationale and contextualises the study much better. Although we do think that felt emotions have received more attention in terms of the emotion structures, but we completely agree that this is kind of beside the point made here; It is much more valuable to say that the context has not been looked at, and this might have simplified/reduced the range of appropriate affects and structures. We have now rewritten the abstract to reflect this crucial shift in the emphasis, and this really helps to portray the present study much better than the previous version. We had to shorten other parts of the abstract, but now it is 200 words and communicates the novel aspects of the study much better. Thank you!

R: In this revision, we also incorporated an observation from a colleague after a conference presentation of this study. The observation relates to the interpretation of the meta-factors in Experiment 3, which bear some striking resemblance to the broad music preference factors (5 factors in the MUSIC model by Rentrow et al., 2011). We point this semblance out and ponder whether the possible underlying reason for the similarity is the contextualised nature of the work done here and how music genres are often also used by listeners. We hope this uncontroversial short addition (108 words, see page 15 of the manuscript) to the interpretations could also stimulate future work on affects and uses of music.

---

## [Editor Report · Decision Letter 2]

25 Oct 2024

What Emotions Does Music Express? Structure of Affect Terms in Music Using Iterative Crowdsourcing Paradigm

PONE-D-24-10867R2

Dear Dr. Eerola,

We’re pleased to inform you that your manuscript has been judged scientifically suitable for publication and will be formally accepted for publication once it meets all outstanding technical requirements.

Kind regards,

Bruno Alejandro Mesz, Ph.D.

Academic Editor

PLOS ONE

---

## [Editor Report · Acceptance letter]

31 Oct 2024

PONE-D-24-10867R2 

PLOS ONE

Dear Dr. Eerola, 

I'm pleased to inform you that your manuscript has been deemed suitable for publication in PLOS ONE. Congratulations! Your manuscript is now being handed over to our production team.

Kind regards, 

on behalf of

Dr. Bruno Alejandro Mesz 

Academic Editor

PLOS ONE